# Immunization against Hepatitis B Surface Antigen (HBsAg) in a Cohort of Nursing Students Two Decades after Vaccination: Surprising Feedback

**DOI:** 10.3390/vaccines8010001

**Published:** 2019-12-19

**Authors:** Maria Gabriella Verso, Claudio Costantino, Francesco Vitale, Emanuele Amodio

**Affiliations:** 1Department of Health Promotion Sciences, Maternal and Infantile Care, Internal Medicine and Medical Specialties “G. D’Alessandro”, Occupational Health Unit, University of Palermo, Piazza Marina, 61, 90133 Palermo, PA, Italy; 2Department of Health Promotion Sciences, Maternal and Infantile Care, Internal Medicine and Medical Specialties “G. D’Alessandro”, Section of Hygiene, University of Palermo, Piazza Marina, 61, 90133 Palermo, PA, Italy; claudio.costantino01@unipa.it (C.C.); francesco.vitale@unipa.it (F.V.); emanuele.amodio@unipa.it (E.A.)

**Keywords:** HBV infection, HBV vaccination, Anti-HBs titer, Healthcare students, work related biological risk

## Abstract

Health-care students can be exposed to biological risks during university training. The persistence of long-term immunogenicity against hepatitis B virus (HBV) was analyzed in a cohort of nursing students two decades after primary vaccination. A total of 520 students were enrolled at the University of Palermo and were evaluated for levels of anti-HBsAg antibodies. The students were examined during the first year of their Degree Course and were checked two years later. All students with anti-HBsAg <10 mIU/mL during their first or third year were boosted within one month. The proportion of students that were vaccinated during adolescence showing anti-HBsAg ≥10 mIU/mL was higher than that observed in students who were vaccinated during infancy (69% versus 31.7%; *p*-value < 0.001). Receiving HBV vaccination at adolescence was significantly associated with a fourfold increased possibility of having anti-HBsAg titers ≥10 mIU/mL (adj-OR = 4.21, 95% CI: 2.43–7.30). Among the students who were checked at the third year and boosted after the first year (*n* = 279), those who were vaccinated during infancy showed a higher percentage of antibody titers <10 mIU/mL (20.3% versus 8.7% among vaccinated during adolescence; *p* < 0.01). This study confirms that HBV vaccination at adolescence might determine a higher long-term persistence of anti-HBsAg titers ≥10 mIU/mL and that anti-HBV booster could increase levels of anti-HBsAg over a relatively short period, especially in subjects who were vaccinated during infancy.

## 1. Introduction

The World Health Organization (WHO) estimates that about 59 million health care workers (HCWs) are exposed to multiple occupational hazards every day, the most common being the risk of exposure to infected patients and/or infectious materials, including body fluids, contaminated medical supplies and equipment, environmental surfaces, or air [1].

In particular, hepatitis B virus (HBV) represents the most transmissible blood-borne virus following percutaneous exposure among healthcare workers [2].

In Italy, according to the national law, healthcare students are considered workers and, therefore, if they are exposed to physical, chemical, biological, or psychological risks, they should be examined by an occupational health physician to evaluate the risks related to their practical activities [3].

Standard precautions, the adoption of enhanced percutaneous injury precautions, and HBV vaccination of HCWs have been demonstrated to consistently reduce the risk of occupational infections and prevent nosocomial transmission of the virus [4]. 

About 95% of the population who are vaccinated against HBV will develop an effective immune response, which can be confirmed by the evaluation of antiHBsAg antibodies (protective level ≥10 mIU/mL) [5]. 

Some studies suggest that the acquired immunity persists for at least 10 years after vaccination with antibody levels ≥10 mIU/mL, however probably not longer if vaccination was performed during the neonatal age [6,7].

In Italy, universal and compulsory HBV vaccination was introduced in 1991 with a two-cohort strategy [8]. In particular, since 1991, HBV vaccination was administered to infants and 12-year old adolescents on a national scale [8]. 

In Italy, among subjects who were vaccinated against HBV from 1991 to 2003, higher immunization levels were observed among those who were vaccinated during adolescence [9].

The 2017 to 2019 Italian National Vaccination Plan strongly recommends the active offer of HBV vaccination to all healthcare professionals [10].

Several researches conducted among students attending healthcare courses highlighted the importance of measuring the immunological response against HBV and of the need for a booster dose in subjects with not protective values, since they have to be considered at higher risk of acquiring HBV infection due to traineeships and/or internships in hospital units [11,12,13,14,15,16].

The main goal of this study was to evaluate persistence of long-term immunogenicity of anti-HBV vaccination and to identify possible predictive factors associated with a stronger immunological response to primary vaccination.

Moreover, a reassessment was carried out during the third year of the study course, two years after the first examination, and levels of antiHBsAg were analyzed and discussed for students that were boosted and those that were not.

## 2. Materials and Methods

From November 2015 to April 2019, a cohort of 520 students attending the nursing schools at the University of Palermo, Italy was evaluated for serum HBsAg, anti-HBs, and anti-HBc. Students were examined during their first and third academic years for evaluating occupational risks. The first-year examination was conducted in 2015/2016, whereas the third-year examination was conducted during 2018/2019.

A standardized medical record including socio-demographic factors (age, gender, country of origin), clinical information (relatives’ diseases and personal remote and proximate pathologies), and previous HBV vaccination was filled-out for each student. Moreover, a medical examination and a blood withdrawal were performed for each student.

Students who were negative for HBsAg and anti-HBc, with titers of anti-HBs lower than 10 mIU/mL, were subsequently boosted and the titer was reassessed after one month.

According to Italian law, a written informed consent was obtained from all the subjects [17].

The study was approved by the Ethical Committee of the University Hospital “P. Giaccone” of Palermo (Protocol number 26/2016 of the 19th of October 2016).

All enrolled nursing students were previously vaccinated against HBV, they were HBsAg negative, and none of them received a vaccination booster before the enrollment.

Among 520 students who were enrolled at the occupational examination for their first year, 449 were further evaluated at a follow-up visit at the third year and 71 were considered lost at follow-up due to changing or dropping-out of the study course.

The flow chart that is reported in Figure 1A,B describes the study population.

All students with a titer ≤10 mIU/mL at the first visit (298 out of 436 vaccinated during infancy, 26 out of 84 vaccinated during adolescence) were boosted within one month.

After the booster, seroprotection rates were similar in students who were vaccinated during adolescence (92% of which 66.6% reporting levels of anti-HBsAg ≥1000 mUI/mL) and in those who were vaccinated during infancy (94.1% of which 45.4% reporting levels of anti-HBsAg ≥1000 mUI/Ml). At the third year of the course, a higher percentage of students with anti-HBsAg <10 required a booster among those who were vaccinated during infancy compared to those who were vaccinated during adolescence, independently from having received a booster dose at the first year of the study course (20% versus 8.1%).

### 2.1. Serological Tests

Serological analyses were performed with commercial chemiluminescence assays (VITROS anti-HBs assay on the Vitros ECI Immunodiagnostic system, Ortho-Clinical Diagnostics, United Kingdom).

In particular, the antibody to the hepatitis B surface antigen (anti-HBs) levels were expressed as mIU/mL. A dynamic range of quantification is 10–1000 mIU/mL. A level of anti-HBs above 10 mIU/mL was considered as protective against HBV infection.

### 2.2. Statistical Analysis

Absolute and relative frequencies were calculated for the categorical (qualitative) variables and normally distributed quantitative variables were summarized by their means (standard deviations). The differences in the categorical variables were analyzed using chi-squared tests (or Fisher’s exact test when appropriate) and the Student t-test for the means.

All the variables that were found to have an association with protective Hepatitis B surface antibody titers (≥10 mIU/mL) at the first visit were included in a multivariate backward stepwise logistic regression model. Crude and adjusted OR with 95% confidence intervals (CIs) were also calculated in the logistic regression model. All information were entered into a database created with Excel 10.0. All data were analyzed using the statistical software package Stata/MP 12.1 (StataCorp LP, College Station, USA).

## 3. Results

The main characteristics of the subjects that were included in the study are shown in Table 1. A total of 520 students (*n* = 436 vaccinated during infancy, 83.8%; *n* = 84 vaccinated during adolescence, 16.2%) underwent an occupational examination during the first year of their degree course. The mean age was 20.5 years (SD±1.6) for students who were vaccinated during infancy and 29.3 years (SD ± 4.9) for students who were vaccinated during adolescence (*p*-value < 0.001). A higher percentage of female students among those that were vaccinated during infancy (66.3% versus 59.5% among students vaccinated during adolescence) was observed.

The mean of years passed from the primary cycle of HBV vaccination was significantly higher among students who were vaccinated during infancy than among students vaccinated during adolescence (20.5 ± 1.6 versus 17.3 ± 4.9; *p*-value < 0.001). Fifty-eight (69%) students who were vaccinated during adolescence showed Hepatitis B surface antibody titers ≥10 mIU/mL, while the prevalence of subjects with anti-HBs titers ≥10 mIU/mL among students vaccinated during infancy was 31.7% (*p*-value < 0.001).

In Table 2, the results of the univariate and multivariate analysis of factors associated with protective levels of anti-HBs at first examination are reported. After adjustment for confounding variables, receiving HBV vaccination during adolescence was significantly associated with increased possibilities of having Hepatitis B surface antibody titers ≥10 mIU/mL (adj-OR = 4.21, 95% CI = 2.43–7.30). In the multivariable analysis, years passed since the first HBV vaccination schedule and gender were not significantly associated with higher anti-HBsAg values.

In Table 3, the serological characteristics of the student cohort that was reassessed for HBV at control examination conducted during the third year of the degree (*n* = 449) were reported. Subjects vaccinated during infancy (*n* = 375) showed a higher percentage of anti-HBsAg <10mUI/mL (20% versus 8.1% among those vaccinated during adolescence; *p* < 0.01). A higher percentage of students with anti-HBsAg <10 mUI/mL was observed, although this was not significant when also considering students boosted (20.3% versus 8.7% among those vaccinated during adolescence) and not boosted after the first visit (19.3% versus 7.9% among those vaccinated during adolescence).

## 4. Discussion

Undoubtedly, during their practical training, students attending a medical degree course, such as nursing students, can be professionally exposed to the risk of contracting infectious diseases as well as HBV infection [2,18].

Several studies conducted on similar populations showed that subjects vaccinated at birth or during adolescence maintain a protective antibody titer in the following 10 years in most cases, while the percentage is reduced 20 years after the primary vaccination schedule, especially if the vaccination was performed during the neonatal age [9,19,20].

In particular, two recently published Italian studies found a lower proportion of subjects with anti-HBs titre <10 mIU/mL than those observed in our cohort (37.7% in Palermo versus 30.8% in Florence and 12% in Rome) [21,22]. These findings are quite different when also considering the time of vaccination and a higher proportion of subjects with non-protective antibody titers was found in our cohort, both considering subjects vaccinated during infancy (68.3% in Palermo versus 51% in Florence and 22.8% in Rome) and subjects vaccinated during adolescence (31% in Palermo versus 12% in Florence and 10% in Rome) [21,22]. Unfortunately, mean age and intervals between primary vaccination and antiHBs titers check seem to be quite similar between the different studies. To date, the reasons for such differences are not clear to us and further investigations are required. On the other hand, in all three settings, vaccination during adolescence elicited higher anti-HBs titres compared to vaccinations performed during infancy.

The different immunological response that was observed among young healthcare professionals who were vaccinated during infancy and during adolescence could be, at least in part, attributable not only to the maturity of the immune response system at the time of the first vaccination cycle completion, but also to the lower dosage contained in the neonatal anti-HBV vaccines compared to some vaccines administered during adolescence [23].

This study seems to confirm these findings and shows that after adjusting for years passed since the first vaccination, statistically significantly higher anti-HBsAg values were observed among nursing students vaccinated during adolescence.

Moreover, the rate of anti-HbsAg antibody titer <10 mIU/mL was surprisingly higher in the cohort of subjects who were vaccinated during infancy that were boosted two years before (*n* = 52, 69.3% of subjects with anti-HBsAg <10 mIU/mL at the third year of the study course), also suggesting that after anti-HBV booster, subjects that are vaccinated during infancy could rapidly decrease their anti-HBsAg levels below 10mIU/mL.

In Italy, several studies have demonstrated a low endemic level of HBV (prevalence of HBsAg in the general population <2%), with an incidence of about 1 case per 100,000 individuals and this result is indubitably attributable to the universal vaccination strategies adopted in 1991 [8,24].

The epidemiology of HBV in developed countries suggests that healthcare workers, and in particular, healthcare students, could have a low risk of contracting the disease, despite not being negligible [25]. For these reasons, to date in Italy, for healthcare professionals as well as for medical residents, students, and trainees, the vaccination against HBV is actively strongly recommended and is free of charge, however there are no mandatory anti-HBV vaccination booster policies [10].

The results of the present study confirm that among the boosted subjects, different responses could be observed and, generally, there was a significant correlation between higher levels of anti-HBs after booster and protective anti-HBs titers after two years.

This correlation was also observed, even though it was not statistically significantly, between high baseline levels of anti-HBs and anti-HBs titers ≥10 mIU/mL after two years. These two correlations were independent from the primary administration of the HBV vaccination schedule.

Future studies that analyze the modification of anti-HBs levels over time for not boosted and boosted healthcare workers could take these results into consideration.

Moreover, the data obtained in this study could also help to further clarify the effect of the HBV vaccination booster, demonstrating a rapid antibody response from the primary vaccination cycle several years later in subjects that were originally responders [25].

Even though to date, a large majority of Italian healthcare students, trainees, medical residents, and workers ≤39 years old have been vaccinated according to the national immunization schedule, a standardized screening for HBV, provided by Occupational Medicine services, it is fundamental not only to identify subjects with low levels of anti-HBsAg or non-responders after primary vaccination cycle, but also for detecting HCWs older than 40-years-old that have not been vaccinated at birth or at 12 years of age.

In this sense, it should be noted that the Italian Legislation recommends the administration of an HBV booster dose for health care workers with anti-HbsAg antibody titer <10 mIU/mL [26].

At first, this recommendation seems to conflict with the World Health Organization position paper on Hepatitis B vaccines, suggesting that subjects with an anti-HBs titer <10 mIU/mL still retain immune memory and therefore, a booster dose in the routine immunization schedule is not recommended [27].

Our results confirm that once the presence of immunological memory in these subjects is demonstrated, regardless of the decline observed in a limited group of boosted subjects two years before, it would no longer make sense to retest them periodically for anti-HBsAg in health surveillance protocols [28,29].

Moreover, despite declining serum levels of anti-HBsAg, international evidences show that vaccine-induced immunity continues to prevent clinical disease or detectable HBV infection [30].

Unfortunately, this study could have some limitations. A first possible limitation is due to the relatively small sample size, which could have reduced the relevance of the statistical tests in finding statistically significant differences. A second limitation is the lack of data about the first vaccination schedule (mostly type of vaccine, etc.). Moreover, the presence of some students lost to laboratory follow-up could have introduced a potential selection bias.

Despite the presence of these possible limitations, our findings confirm the expected reduction of anti-HBsAg two decades after the HBV primary vaccination schedule, which was more evident in subjects who were vaccinated during infancy due to a probable immaturity of the immune system at that time or to the use of a vaccine with lower antigenic content [31].

## 5. Conclusions

This study confirms that HBV vaccination during adolescence might determine a higher long-term persistence of anti-HBsAg titers ≥10mIU/mL and that HBV booster could increase levels of anti-HBsAg over a relatively short period, especially in subjects who were vaccinated during infancy.

Moreover, more than two-thirds of the subjects vaccinated during infancy showed levels of anti-HBsAg <10 mIU/mL at the first visit and after having developed a valid immunity one month after the booster dose, among 20% of students boosted two years earlier, anti-HBsAg levels were again <10 mIU/mL at the control visit (third year).

It is highly likely that when encountering the natural HBV virus, subjects who were vaccinated or boosted that showed anti-HBsAg levels <10 mIU/mL at the first or third visit will produce a valid antibody reaction that is capable of antagonizing the infection.

These findings support the recommendation by the regulatory Health Agencies to screen and administer a booster dose for first time individuals belonging to high-risk groups, such as professionals employed in health-care services, and in future, could represent a suggestion for the management of health-care workers vaccinated against HBV during infancy or adolescence according to their anti-HBs immunization status [32].

In future, it could be useful to increase the number of subjects enrolled in the study to confirm and increase the generalizability of our results, identifying other variables that could play a further role in determining the seroconversion rate and perdurance of antibody levels.

## Figures and Tables

**Figure 1 vaccines-08-00001-f001:**
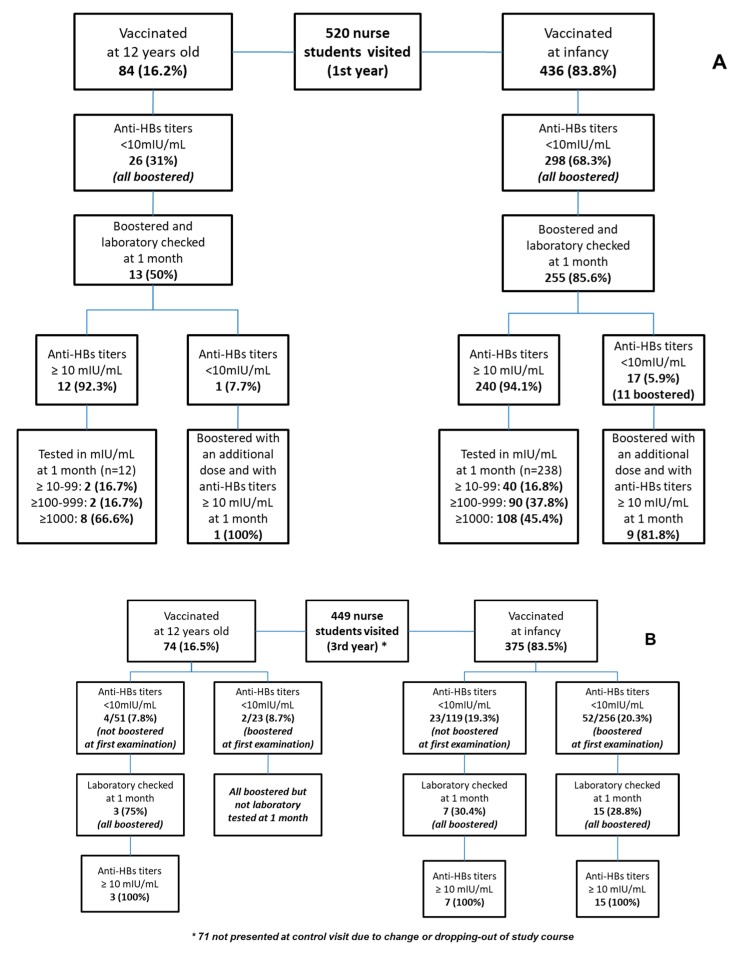
(**A**) and (**B**). Students enrolled at occupational examinations conducted during the 1st (*n* = 520) and 3rd year (*n* = 449) of the nursing study course.

**Table 1 vaccines-08-00001-t001:** General characteristics of the nursing students enrolled during the 1st year of the study course (*n* = 520).

*n* = 520	Vaccination during Infancy (*n* = 436)	Vaccination during Adolescence (*n* = 84)	*p*-Value
Mean age ± DS	20.5 ± 1.6	29.3 ± 4.9	<0.001
Gender, *n* (%)			
Male	147 (33.7)	34 (40.5)	0.14
Female	289 (66.3)	50 (59.5)	
Years from vaccination, mean ± DS	20.5 ± 1.6	17.3 ± 4.9	<0.001
Anti-HBs titers, *n* (%)			
<10 mIU/mL	298 (68.3)	26 (31.0)	<0.001
≥10 mIU/mL	138 (31.7)	58 (69.0)	

**Table 2 vaccines-08-00001-t002:** Factors associated with protective Hepatitis B surface antibody titers (≥10 mIU/mL) at the first year occupational examination among nursing students (*n* = 520).

	Crude OR	95% CI	*p*-Value	AdjOR	95%CI	*p*-Value
**Years since HBV vaccination**	0.87	0.81–0.93	<0.01	0.96	0.89–1.03	0.25
Gender						
Male	ref		0.60	ref		0.80
Female	0.90	0.62–1.31		0.95	0.64–1.40	
HBV vaccination timing						
vaccinated during infancy	ref		<0.001	ref		<0.001
vaccinated during adolescence	4.81	2.90–7.97		4.21	2.43–7.30	

**Table 3 vaccines-08-00001-t003:** Anti-HBs titers observed among nursing students during their occupational examination during the third year of their study course (*n* = 449).

	Vaccination during Infancy *n* (%)	Vaccination during Adolescence *n* (%)	*p*-Value
**Anti-HBs titers at the control visit (third year), (*n* = 449)**			
<10 mIU/mL	75 (20.0)	6 (8.1)	<0.01
≥10 mIU/mL	300 (80.0)	68 (91.9)	
**Anti-HBs titers at the control visit (third year) among students not boosted after their first visit, (*n* = 170)**			
<10 mIU/mL	23 (19.3)	4 (7.9)	0.06
≥10 mIU/mL	96 (80.7)	47 (92.1)	
**Anti-HBs titers at the control visit (third year) among students boosted after their first visit, (*n* = 279)**			
<10 mIU/mL	52 (20.3)	2 (8.7)	0.18
≥10 mIU/mL	204 (79.7)	21 (91.3)	

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
