# Peer review of "Immunization against Hepatitis B Surface Antigen (HBsAg) in a Cohort of Nursing Students Two Decades after Vaccination: Surprising Feedback"

_vaccines, 2019, doi:10.3390/vaccines8010001_

Round 1
Reviewer 1 Report
Dear Editor,
I read the manuscript entitled "HBV immunization against Hepatitis B Surface Antigen (HBsAg) in a cohort of nursing students two decades after vaccination: a surprising feedback.". The study presented by Verso and colleagues is an interesting study performed in an endemic area of HBV infection of South Italy.
The study is well described and performed with a young population previously vaccinated during adolescence versus infancy. The cohort number is adequate and statistical analysis is correct. The results are interesting and deserve to be published.
Reviewer 2 Report
The study by Verso et al. Describes the anti-HBV vaccination status in 1 cohort of nursing student who have been vaccinated either at infancy or at adolescence, comparing the 2 situations in terms of immunization persistence, HBsAg titers and efficacy of booster vaccination in subjects with <10 mIU/ml. This study is certainly not original, as a wealth of similar studies with bigger populations have been published on this topic, many of which from Italy. By the way, the authors fail to cite in particular 2 of those, very similar to the one described in this manuscript, respectively from Florence (10.1080/21645515.2017.1398297) and Rome (10.3390/ijerph16091515,), the former reporting results partially contrasting with those of the current manuscript, which should be commented on. The only novelty in the current manuscript, which might support (although with low priority) its publication, is the 2 year titer follow-up after the booster dose. However there are a few important issues concerning this study (design and results reporting and discussing) that should be addressed:
In the 3° year testing, the cohort should be divided in 4 groups (infancy/adolescence and boosted/ not boosted) for all the analyses, otherwise the data will be confounding. So figure 1B, table 3 and table 4 should be rewritten, as the relative description in the text. The sentence from line 161 “in Particular…” describes useless analyses which should be eliminated. The discussion should be shortened, other results from Italy should be included in the discussion, and more attention should be dedicated to the 2 years follow-up after the boost in the first year. The conclusions should also be focused on this point. The bibliography should be updated including the studies from Florence and Rome. English style needs some revisionsAuthor Response
Please see the attachment

Reviewer 3 Report
COMMENTS AND SUGGESTIONS
English language editing is necessary e.g. line 44 undergoing to HBV – to is redundant, line 56 active not actively, line 59 of measuring the immunological response, line 71 against HBs or just anti-HBs, line 76 student not students, line 78 against HBc or just anti-HBc, line 83 were HBsAg…, line 85 From 520 etc.
Line 76-77 What is a personal objective exam?
Lines 89-95 belong to the result section.
In figure 1A, 4th box on the right (vaccinated in infancy, boostered and responded) anti-HBs titers>10 IU/mL is it 240 0R 238? The numbers do not add up
Lines 152-154 the meaning is not clear please rephrase.
Lines 161-167 It would be preferable to keep it simple with main point that protective immunity was found mainly in boostered students and those with high anti-HBs at first visit (if I understand it correctly)
Since Table 4 does not contain those students that were not checked after booster at 1st year it would be interesting to mention their status at year 3
Round 2
Reviewer 2 Report
The present form of the manuscript is now acceptable for publication.
Author Response
In attachment a point-by-point response to the reviewer's comment
